# CD69 Signaling in Eosinophils Induces IL-10 Production and Apoptosis via the Erk1/2 and JNK Pathways, Respectively

**DOI:** 10.3390/biom14030360

**Published:** 2024-03-18

**Authors:** Dan Van Bui, Linh Manh Nguyen, Akira Kanda, Hanh Hong Chu, Nhi Kieu Thi Le, Yasutaka Yun, Yoshiki Kobayashi, Kensuke Suzuki, Akitoshi Mitani, Akihiro Shimamura, Kenta Fukui, Shunsuke Sawada, David Dombrowicz, Hiroshi Iwai

**Affiliations:** 1Department of Otolaryngology, Head and Neck Surgery, Kansai Medical University, Osaka 573-1010, Japandr.linh1992@gmail.com (L.M.N.); suzukken@hirakata.kmu.ac.jp (K.S.); ken0630bbt@gmail.com (K.F.); iwai@hirakata.kmu.ac.jp (H.I.); 2Allergy and Clinical Immunology Department, Hanoi Medical University, Hanoi 115000, Vietnam; 3Allergy, Immunology and Dermatology Department, E Hospital, Hanoi 122000, Vietnam; 4Laboratory Medicine Center, Kansai Medical University Hospital, Osaka 573-1010, Japan; 5Allergy Center, Kansai Medical University Hospital, Osaka 573-1010, Japan; 6Allergy, Immunology and Rheumatology Department, National Children Hospital, Hanoi 115000, Vietnam; 7Dentistry and Oral Surgery and Care Center, Kansai Medical University Hospital, Osaka 573-1010, Japan; sawadash@hirakata.kmu.ac.jp; 8University of Lille, Inserm, CHU Lille, Institut Pasteur de Lille, 59000 Lille, France; david.dombrowicz@pasteur-lille.fr

**Keywords:** asthma, CD69, crosslinking, Erk1/2, eosinophil, IL-10, JNK, type 2 inflammation, survival

## Abstract

Introduction: Eosinophils contribute to the pathogenesis of allergic diseases, including asthma, allergic rhinitis, and atopic dermatitis. We previously reported that human tissue eosinophils have high CD69 expression compared to blood eosinophils, and its expression is correlated with disease severity and the number of infiltrated eosinophils. However, biological CD69 signaling activity in eosinophils remains unclear. Methods: CD69 expression on lung tissue eosinophils obtained from mice with ovalbumin-induced asthma was measured using flow cytometry. CD69 crosslinking was performed on eosinophils purified from the spleen of IL-5 transgenic mice to investigate CD69 signaling and its function in eosinophils. Then, qPCR, Western blot, enzyme-linked immunosorbent assay, and survival assay results were analyzed. Results: Surface CD69 expression on lung tissue eosinophils in the asthma mice model was 2.91% ± 0.76%, whereas no expression was detected in the healthy group. CD69-expressed eosinophils intrinsically have an upregulation of IL-10 mRNA expression. Moreover, CD69 crosslinking induced further pronounced IL-10 production and apoptosis; these responses were mediated via the Erk1/2 and JNK pathways, respectively. Conclusions: Our results suggested that CD69^+^ eosinophils play an immunoregulator role in type 2 inflammation, whereas activated tissue eosinophils contribute to the pathogenesis of asthma.

## 1. Introduction

Eosinophils not only play an important role in homeostasis maintenance and host defense against helminth infection but also contribute to the pathogenesis of type 2 inflammation, such as asthma, allergic rhinitis, and atopic dermatitis [1,2,3,4]. They exert their biological effects by releasing eosinophil-derived cytotoxic granules, protein, cytokines, chemokines, and lipid mediators [5,6,7]. CCR3, FcεRI, CD9, CD11b, CD11c, CD13, CD18, CD25, CD63, CD69, CD123, CD125, and GM-SCFR have been known as activation markers of eosinophils in the tissue [1,6,8].

CD69, a very early activation antigen, regulates the differentiation of regulatory T cells (Treg) and the secretion of cytokines, such as interferon-gamma (IFN-γ), IL-17, and IL-22 [9]. This receptor is upregulated on activated T cells, natural killer (NK) cells, B lymphocytes, neutrophils, and eosinophils [9,10]. The increase in CD69 on leukocytes at the inflamed site is observed in the atopic dermatitis and asthma mouse models [11]. However, it has been reported that highly expressed CD69 T cells in lung tissue have a conflicting effect, such as acting as both a promoter and regulator, in the immune allergic response [11,12,13].

Recent papers showed that CD69 expression on eosinophils has a correlation with the development of type 2 immune response [10,14,15]. Indeed, we previously reported that activated tissue eosinophils in nasal polyps obtained from eosinophilic chronic rhinosinusitis (ECRS) patients with asthma highly expressed CD69 [10,14,15]. This upregulation was correlated with disease severity as well as with the number of infiltrated eosinophils in the tissues, suggesting that an increase in CD69 expression in eosinophils is an important marker of the degree of exacerbation in ECRS patients with asthma [10,14,15]. However, little is known about the biological activity of CD69 signaling on eosinophils.

To investigate the functional role and signaling of the CD69 receptor in vitro, a crosslinking method is commonly performed. When CD69 undergoes crosslinking on T cells, it upregulates IL-2 and IFN-γ mRNA expression, with an increase in Ca^2+^ influx [16], as well as induces the ability to kill target cells as a cytolytic function [17]. In a study using CD69-deficient mice, Martín et al. showed that CD69 signaling induced T cell-differentiated T helper (Th) 1 and Th2 but not Th17, suggesting that it contributed to the intrinsic modulation of T cell differentiation during immunoinflammatory processes [18]. Moreover, serotonin degranulation and cytotoxic activity were observed in NK cells following CD69 crosslinking [19]. By contrast, Yu et al. reported that CD69 crosslinking induces an increase in IL-10 production from Foxp3^+^ regulatory T cells, thereby playing an immunosuppressive role and attenuating colitis in animal models [20]. Together, these results suggested that the functional role of CD69 depends on the cell type and/or immune environment status.

IL-10 is produced by monocytes, epithelial cells, mast cells, macrophages, regulatory T cells, regulatory B cells, and eosinophils [21,22,23]. Although IL-10 is a potent regulatory cytokine, it is known to upregulate in the severe and/or chronic type 2 inflammation. Similarly, we previously found that IL-10 as well as Th2 cytokines were increased in the human eosinophils in nasal polyps [9,10]. In antigen-presenting cells, including macrophages and dendritic cells, the stimulation of Toll-like receptor (TLR) 2, TLR4, and TLR9 significantly increases IL-10 release following the augmentation of nuclear factor-κB via the phosphorylation of the extracellular signal-regulated kinase (Erk) 1/2 and p38 [24,25]. Saraiva et al. demonstrated that the sustainable enhancement of Erk1/2 phosphorylation in the Th1 cells induced IL-10 production following stimulation with high doses of antigen and IL-12 [26] in the differentiated Th subsets. Moreover, they observed that Th2 and Th17 cells stimulated with anti-CD3/CD28 in vitro significantly increased IL-10 production and that this effect was reversed by the selective noncompetitive MEK inhibitor PD184352 [26]. Although eosinophils intrinsically have IL-10 and secrete it following IgA stimulation [27], the mechanisms underlying the regulation of IL-10 secretion in eosinophils remain to be fully elucidated.

We had previously reported that high-concentration CD69 crosslinking immediately released EPO granules from eosinophils [10,14,15], whereas Walsh et al. showed that this crosslinking induced apoptosis in human eosinophils [28]. By producing IL-10 and IFNγ, eosinophils also regulate the Th1/Th2 balance [6]. Thus, the functional role of CD69 expressed on eosinophils by CD69 ligation remains unclear. Here, we first measured the percentage of CD69 expressed on eosinophils in ovalbumin (OVA)-induced asthma mouse models, following which we investigated whether CD69 crosslinking on eosinophils induces cytokine release and apoptosis and results in the activation of signaling pathways.

## 2. Methods

### 2.1. Animals

Female BALB/c mice (aged 6–8 weeks) were purchased from Shimizu Experimental Materials (Kyoto, Japan). The IL-5 transgenic mice on a BALB/c background were provided by Dr. D. Dombrowicz (Institut Pasteur de Lille, Lille, France). All mice were housed in a specific pathogen-free animal facility at an appropriate temperature and humidity with a regular 12 h light/dark cycle. The Animal Care and Use Committee of Kansai Medical University (18-082) approved all experimental procedures used in this study.

### 2.2. OVA-Induced Asthma Model

An intraperitoneal injection of 50 µg OVA (Sigma-Aldrich, St. Louis, MO, USA) emulsified in 1 mg aluminum hydroxide (Thermo Fisher Scientific, Waltham, MA, USA) was administered on days 0 and 14 for sensitizing the mice; thereafter, the mice were challenged with 5% OVA (15 µL each nostril) intranasally daily from day 21 to 25. Mice treated with phosphate-buffered saline (PBS) were used as controls. Mice were sacrificed on day 26 to collect peripheral blood and lung tissues for analysis (Appendix A). A detailed protocol of lung cell preparation and flow cytometric analysis is shown in the online Appendix A.

### 2.3. Eosinophil Purification

It is difficult to purify CD69+ eosinophils from the lung tissue of naïve mouse as well as human samples; for instance, there are no more than 100,000 cells in a 1 g nasal polyp. As a result, we performed an in vitro experiment using hypereosinophilia mice, named IL-5 transgenic mice (Tg) [10]. Eosinophils were purified from the spleen of IL-5 Tg mice with eosinophilia using negative selection, as reported in a previous study [10,14,15] (described in the online Appendix A).

### 2.4. CD69 Crosslinking on Eosinophils

CD69 crosslinking on eosinophils was performed according to a previous protocol [10,14,15] for investigating the functional role of CD69 expressed on eosinophils. Briefly, purified eosinophils were incubated with 10 ng, 100 ng, 1 μg, or 10 μg/mL of anti-mouse CD69 mAb (Novus Biologicals, Centennial, CO, USA) or with Armenian hamster IgG (BioLegend, San Diego, CA, USA) as the control at 4 °C for 30 min. Then, cells were transferred into a 24-well plate coated with 20 μg/mL of goat anti-Armenian hamster IgG as a secondary antibody (Jackson ImmunoResearch, West Grove, PA, USA).

### 2.5. Functional Assay of Eosinophils

Quantitative polymerase chain reaction (qPCR) was performed using a QuantiTect SYBR Green PCR Kit (Qiagen, Hilden, Germany) with the Rotor-Gene Q HRM cycler (Qiagen, Hilden, Germany). The primers used for reverse transcription PCR are listed in Appendix A. ΔΔCt was calculated to determine the relative expression normalized to GAPDH. Protein levels for IL-10 and IL-5 in the culture supernatants were measured using enzyme-linked immunosorbent assay (ELISA) kits from RayBiotech (Peachtree Corners, GA, USA) and R&D System (Minneapolis, MN, USA), respectively, according to the manufacturers’ instructions. Survival assays were analyzed via double negative staining with annexin V and 7-aminoactinomycin D (BD Pharmingen, San Diego, CA, USA) using flowcytometric analysis (BD Biosciences, Franklin Lakes, NJ, USA).

### 2.6. Singal Transduction

To investigate signal transduction in eosinophils following CD69 crosslinking, Western blot analysis was performed as described in the online Appendix A. In the experiments for cell signaling pathway, eosinophils were incubated with or without the Erk1/2 phosphorylation inhibitor PD98059 (MEK inhibitor) (Cell Signaling Technology, Danvers, MA, USA) or with or without the JNK inhibitor SP600125 (Abcam, Cambridge, UK) at 37 °C for 1 h before treatment with a secondary antibody. Then, ELISA and survival assay were performed.

### 2.7. Statistical Analysis

Data are presented as means ± standard errors of the mean (SEMs). Statistical significance was determined using a two-way ANOVA (analysis of variance) and Dunnett’s test with GraphPad Prism version 8 software (GraphPad, San Diego, CA, USA). The threshold of significance was assumed to be a *p* value of <0.05 for all tests.

## 3. Results

### 3.1. CD69 Expression and Characteristics of CD69^+^ Eosinophils

Flow cytometry analysis was performed to investigate the percentage of CD69 expression. Eosinophils in the lung tissue were gated as CD45^+^CD11b^Hi^Gr-1^Int^SiglecF^Hi^ (Appendix A). CD69 expression of 2.91% ± 0.76% was observed on lung tissue eosinophils obtained from mice with OVA-induced asthma, whereas no expression was detected in the peripheral blood or in the PBS-treated group (Figure 1A). Similar results were obtained using immunofluorescence staining (Figure 1B).

Since it is difficult to obtain enough eosinophils for performing an in vitro experiment, we used IL-5 Tg mice with hypereosinophilia. Eosinophils from the naïve spleen of IL-5 Tg mice gated as CD19^−^CD90.2^−^Gr-1^Int^SiglecF^Hi^ (Appendix A) exhibited a surface CD69 expression of 15.96% ± 0.39%, whereas no expression was detected in peripheral eosinophils (Figure 1C). Moreover, this expression was also observed using immunofluorescence staining (Figure 1D). To study the characteristic features of CD69^+^ mouse eosinophils, the mRNA expression of transcription factors Th1, Th2, and Th17 and the regulation of anti-inflammatory cytokines such as IL-10 and transforming growth factor beta (TGF-β) were analyzed in the CD69^+^ and CD69^−^ eosinophils from the naïve spleen of IL-5 Tg mouse, purified using a cell sorter (over 99.9% purity). Only the upregulation of IL-10 mRNA expression was significantly observed in the CD69^+^ eosinophils compared to CD69^−^ eosinophils (Figure 1E).

### 3.2. Cytokine Profiling by CD69 Crosslinking on Eosinophils

To investigate the functional role of CD69 on eosinophils, CD69 crosslinking was performed in the eosinophils purified from the spleen of IL-5 Tg mice. We found that only the upregulation of IL-10 mRNA expression was significantly induced by 10 μg/mL CD69 crosslinking at 1 and 3 h of incubation (Figure 2A). However, no alterations were observed in the gene expressions of INF-γ, T-bet, IL-4, IL-13, GATA3, IL-17, RAR-related orphan receptor gamma, and TGF-β (Figure 2A). This significant increase in IL-10 in the mRNA and protein levels were at least observed from 10 ng/mL CD69 crosslinking for 3 h incubation (Figure 2B,C).

### 3.3. Phosphorylation of Erk1/2 and JNK by CD69 Crosslinking on Eosinophils

We found that CD69 crosslinking induced the phosphorylation of Erk1/2 and JNK but not that of STAT5 and Jak3 (Figure 2D). To investigate whether Erk1/2 inhibitor treatment suppresses IL-10 production induced by CD69 crosslinking, treatment with PD98059 (MEK inhibitor), an Erk1/2 phosphorylation inhibitor, was performed prior to CD69 crosslinking on eosinophils, resulting in the attenuation of Erk1/2 phosphorylation (Appendix A) and, consequently, decreasing the production of IL-10 (Figure 2E). Moreover, we observed that CD69 crosslinking decreased eosinophil survival and pretreatment with SP600125, a JNK inhibitor, and partially reversed eosinophil death by CD69 crosslinking (Figure 2F).

## 4. Discussion

In the present study, we observed an increase in CD69 expression in lung tissue eosinophils obtained from OVA-induced asthma mice models compared with healthy as well as peripheral blood eosinophils. CD69-expressing eosinophils possessed the potential to intrinsically increase IL-10 expression. Moreover, signaling through CD69 crosslinking enhanced further IL-10 production by Erk1/2 phosphorylation and decreased eosinophil survival through the JNK pathway.

Ickrath et al. reported that CD4^+^ T cells from the nasal polyps of patients with ECRS exhibited a surface CD69 expression of 38.25% ± 14.23%; however, this expression was not detected in peripheral blood T cells [29]. Similarly, in our previous paper, we showed a high CD69 expression in activated human eosinophils in nasal polyp tissues obtained from patients with ECRS but none in the peripheral blood [10,14,15]. Here, by contrast, this expression was lower compared with the human-activated eosinophils at the inflamed site in OVA-induced asthma mouse models. However, we observed that CD69 expression in mouse eosinophils obtained from the spleen of IL-5 Tg mice was higher than that in the eosinophils obtained from the lung tissues of OVA-induced mice. Urasaki et al. showed that stimulation isolated eosinophils from the peripheral blood of both patients with hypereosinophilic syndrome and healthy donors with IL-5, thereby significantly inducing CD69 expression. This suggested that IL-5 plays a crucial role in CD69 upregulation in type 2 inflammation [30]. Therefore, this conflict in the mouse model for the proportion of CD69 expression might contribute to the local concentration of IL-5. Indeed, we observed that the IL-5 concentration in the supernatant of splenic cells from IL-5 Tg mice was higher than that of lung cells from the OVA-induced asthma model (Appendix A).

Although eosinophils are an important player in the pathogenesis of asthma exacerbation, they secrete IL-10, suggesting a subtype of eosinophils as a regulator at the inflamed site [27]. Huang et al. reported that eosinophil-derived IL-10 supports helminth infection as an immunoregulatory eosinophil function, resulting in the enhancement of parasite survival [31]. Moreover, Nakajima et al. showed that IL-10 mRNA was constitutively expressed in human eosinophils isolated from healthy volunteers and that IL-10 production was increased by exogenous human recombinant IL-5, thereby suggesting that IL-5 indicates a potential for IL-10 upregulation from eosinophils [32]. Foerster et al. showed that the ligation-specific antibody to CD69 induced eosinophil apoptosis, thereby supporting our results, and CD69^+^ eosinophils induced by GM-CSF prolonged their survival in comparison with CD69^−^ eosinophils [33]. Similarly, Walsh et al. showed that apoptotic human eosinophils were induced by CD69 crosslinking performed with a 48 h incubation [28]. Together, these data suggest that the intrinsic role of CD69 signaling in eosinophils plays an immunoregulatory role via the increase in IL-10 production and cell death in the type 2 inflammation.

Martín et al. reported that CD69 crosslinking to T cells mediated Jak3/STAT5 phosphorylation for CD69 signal transduction, thereby resulting in the promotion of Treg differentiation [18]. Zingoni et al. showed that this signaling in the NK cells induced serotonin release and upregulated the lysis against P815 FcγR^+^ target cells via Erk1/2 phosphorylation [19]. Additionally, the signaling pathway of CD69 expressed on activated neutrophils remains unknown [34]. IL-10 production and eosinophil death enhancement stimulated by CD69 crosslinking on eosinophils were found to be induced by Erk1/2 and JNK phosphorylation, respectively, suggesting that the signaling pathway of CD69 in eosinophils was similar to that in NK cells.

Although we found a functional role of CD69 on eosinophils through forced stimulation, such as the use of CD69 crosslinking, the natural ligand functions remain unclear. Currently, there are two possible ligands to CD69 expressed on eosinophils Gal-1 and Myl9/12 [16,35,36]. Gal-1 stimulation increases IL-10 and decreases IFN-γ production in human peripheral blood mononuclear cells, resulting in an immunosuppressive effect [37]. Conversely, Myl9/12 contributes to the development of pathogenetic inflammation, such as that in asthma, ECRS, and bowel disease [36,38,39]. Further studies are required to clarify the mechanisms of conflict regarding the biological function and signal transduction by natural ligands, although this conflict might result from ligand affinity.

In conclusion, we found an increase in IL-10 in CD69^+^ eosinophils, and CD69 crosslinking on eosinophils induced a further pronounced IL-10 production through the Erk1/2 pathway and increased death via the JNK signaling pathway. Our data suggest a functional role of CD69 expression on eosinophils as an immunosuppressive player in type 2 inflammation.

## Figures and Tables

**Figure 1 biomolecules-14-00360-f001:**
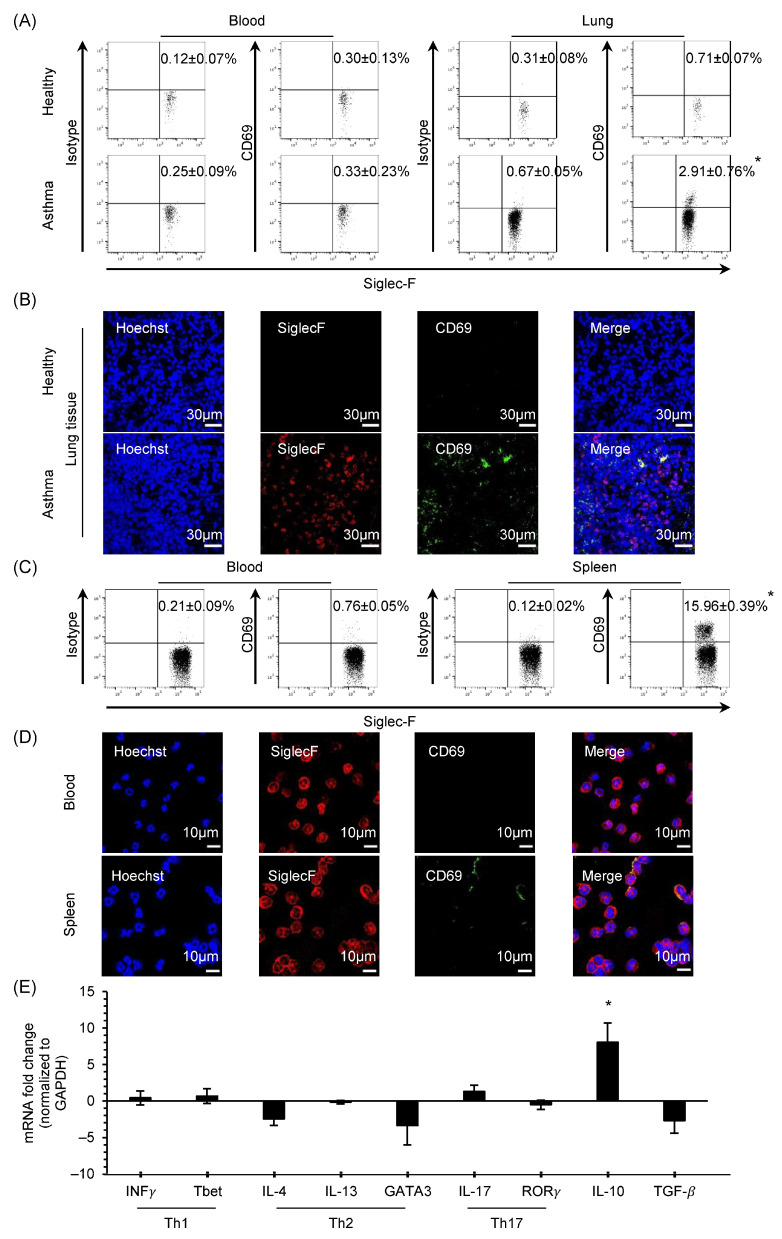
CD69 expression and cytokines profiling of mouse eosinophils. (**A**) Surface expression of isotype IgG and CD69 on peripheral blood and lung eosinophils in the healthy and asthmatic mouse models using flow cytometry (*n* = 6). (**B**) Immunofluorescence staining of healthy and asthmatic lung tissue sections stained with antibodies, with corresponding isotypes IgG, Siglec-F, and CD69. (**C**) Surface CD69 expression on eosinophils from peripheral blood and the spleen in naive IL-5 transgenic (Tg) mice (*n* = 5 for each group). (**D**) CD69 expression observed using immunofluorescence staining in the eosinophils purified from the blood and spleen of naive IL-5Tg mice. (**E**) qPCR analysis of Th1, Th2, and Th17 cytokines; transcription factors; IL-10; and TGF-β mRNA expression in the CD69^+^ eosinophils purified using a cell sorter. Relative gene expressions were calculated with ΔΔCt method using sorted CD69^−^ eosinophils as the reference (*n* = 4 for each group). Data are expressed as mean ± SEM. * indicates a significant difference compared with healthy mice, blood, or CD69^−^ eosinophils (*p* < 0.05).

**Figure 2 biomolecules-14-00360-f002:**
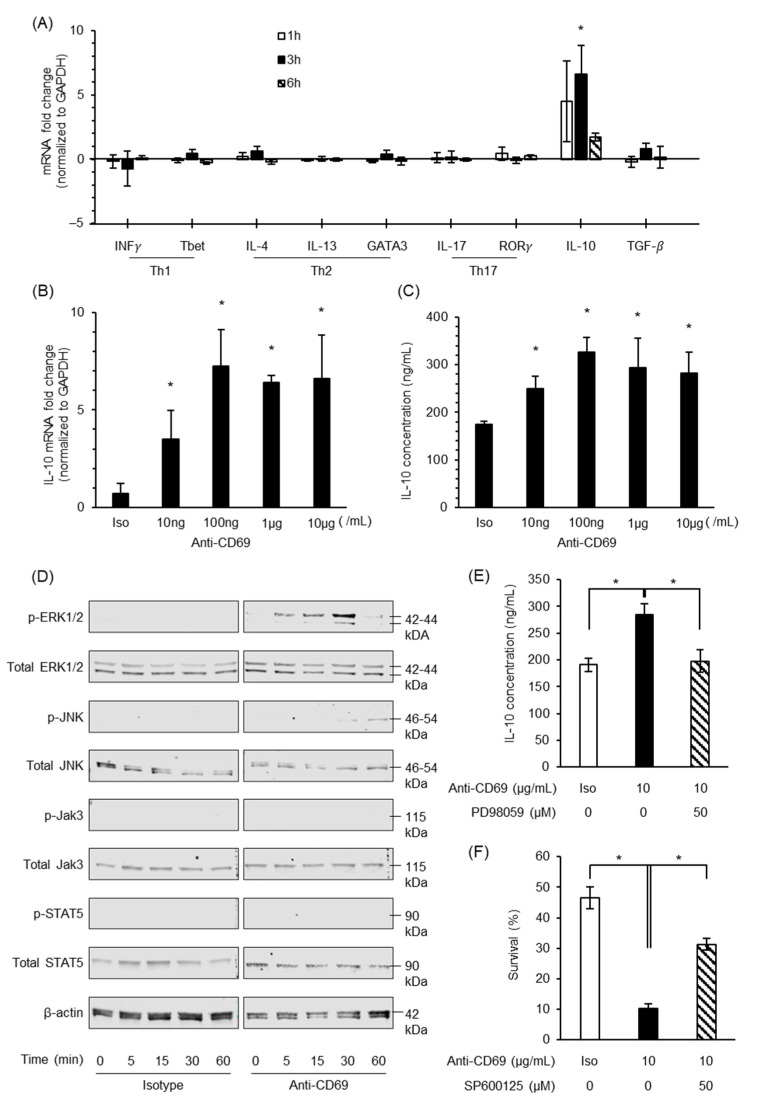
Functional role and signaling pathway of CD69 on eosinophils following CD69 crosslinking. (**A**) qPCR analysis of the Th1, Th2, and Th17 cytokines, transcription factors, IL-10, and TGF-β mRNA expression at 1, 3, and 6 h after 10 μg/mL CD69 crosslinking on eosinophils. The mRNA expressions on CD69^+^ eosinophils were compared with those on non-stimulated eosinophils (isotype), which were considered as reference (*n* = 3 for each group). (**B**,**C**) mRNA expression and protein level of IL-10 after corresponding isotype IgG, 10 ng, 100 ng, 1 μg, or 10 μg/mL CD69 crosslinking on eosinophils for 3 h and 24 h, respectively (*n* = 4 for each group). * indicates a significant difference from isotype group (*p* < 0.05). (**D**) Phosphorylation and total Erk1/2, STAT5, Jak3, and JNK protein and β-actin expression in eosinophils purified from IL-5 Tg mouse following crosslinking by 10 μg/mL CD69 antibody or matched control via Western blot analysis. IL-10 secretion (**E**) and % survival of eosinophils (**F**) following 10 μg/mL CD69 crosslinking on eosinophils purified from the spleen of IL-5 Tg mice for 24 h culture in the absence (positive control group) or presence of 50 μM PD98059 (*n* = 4) and 50 µM SP600125 (*n* = 4), respectively. Data are expressed as mean ± SEM. * indicates a significant difference from 10 μg/mL anti-CD69 stimulation without inhibitor treatment group (positive control group) using a two-way ANOVA (*p* < 0.05). Original figures can be found in Appendix A.

## Data Availability

The data presented in this study are available on reasonable request from the corresponding author due to ethical reasons.

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
