# Peer review of "CD69 Signaling in Eosinophils Induces IL-10 Production and Apoptosis via the Erk1/2 and JNK Pathways, Respectively"

_biomolecules, 2024, doi:10.3390/biom14030360_

Round 1

Reviewer 1 Report

Comments and Suggestions for Authors

The ms. by Van Bui et al “ CD69 signaling on eosinophils induces IL-10 production and apoptosis via the Erk1/2 and JNK pathways, respectively” describes a new immunosuppressive role of CD69 on eosinophils in the asthma context.

The paper is interesting, but several concerns have to be addressed before its consideration for publication.

1)    The authors study OVA-induced asthma model inducing a 2.9% upregulated CD69+eosinophils and  IL-5Tg mice model carrying hypereosinophilia and stimulated by CD69 crosslinking with anti-CD69 Ab. Only in the latter, with 16% CD69+eosinophil induction at spleen level, the  CD69 signaling pathway is studied and it is used to draw the conclusion of the involvement of IL-10 production and eosinophil death increase stimulated by CD69 crosslinking and involving Erk ½ and JNK phosphorylation. How can the involvement of this signaling pathway be translated to the asthma model where only 2.9% upregulated eosinophils are recovered? Explanations and further data should be provided on this topic.

2)     CD69 is triggered through forced stimulation, by antibody crosslinking, and downstream signaling is analyzed. It would be interesting to provide data also in a more closely related in vivo setting, i.e. by using a natural ligand stimulus.

3)    the CD69 immunofluorescence data on spleen is not convincing as presented (Fig 1D), since a barely detectable CD69 fluorescence is shown, and is not in accordance with 16% positive cells as resulting from FACS analysis (Fig 1C). Please address this point.

4)     IL-10 concentration data is badly represented (Fig 2E). The statistic and significance on Iso is versus what?? Same question for the histogram in Fig 2F. The significance should be on anti-CD69 10 mg/mL condition? Please represent more clearly the condition analyzed for statistics.

5)    In the methods section the methodology of CD69+ eosinophils sorting is lacking

6)    A final graphical abstract with models and CD69-pathways engaged would help to convey the overall message.

Comments on the Quality of English Language

overall the quality of English language is good and minor editing is required

Author Response

Dear Reviewer 1

Thank you very much for your reviewing and comments. For your comments and suggestion, please see attached file.

Sincerely,
Prof. Akira Kanda, MD, PhD

Reviewer 2 Report

Comments and Suggestions for Authors

This brief report by Van Bui et al. demonstrates CD69 expression on subsets of (presumably activated) mouse eosinophils in tissues and shows that these cells upregulate IL-10 at the mRNA level without intervention and further augment IL-10 at both the mRNA and protein levels following treatment with anti-CD69 antibody. This effect is found to correlate with and be dependent on ERK1/2 phosphorylation using a pharmacologic inhibitor. JNK is also phosphorylated, and use of a JNK inhibitor, SP600125, impedes cell death induction in response to anti-CD69 antibody treatment. These are interesting findings that help clarify the role of CD69 on eosinophils. There remain several issues that are addressed in the following specific comments:

1.       A previous study by this group showed that eosinophil peroxidase is released in response to high concentrations of anti-CD69. This outcome appears to be at odds with the outcomes described in this manuscript, and some discussion regarding this apparent discrepancy would be valuable. Furthermore, at high concentrations, antibodies may bind their targets monomerically and impede receptor aggregation, and the effects on IL-10 expression appear to be induced with much lower concentrations of anti-CD69. Is there any evidence that these different outcomes are due to crosslinking or monomeric binding? Does a secondary antibody enhance or impede a particular outcome of antibody treatment and was the highest concentration of anti-CD69 antibody ever used to measure effects on IL-10 expression?

2.       Presumably, the inhibitors PD98059 and SP600125 were used in assays that measure both IL-10 secretion and cell viability and effects were only observed in one assay or the other. It would be helpful to present the negative data to support the conclusion that these are distinct signaling pathways. Please also note if the treatment controls also used the same diluent as the pharmacologic inhibitors (DMSO, perhaps), as DMSO itself can exert some biological effects at higher concentrations.

3.       There does not appear to be any cell viability data prior to panel showing inhibition of this outcome with sp600125. The inclusion of flow plots or anti-CD69 antibody concentration dependence would be helpful to interpret the data.

4.       It is this reviewer’s understanding that PD98059 acts directly on MEK, which is an upstream activator of ERK1/2, and is not directly an inhibitor of ERK1/2 activity.

Comments on the Quality of English Language

Overall, the manuscript is well written and organized. There are, however, several instances of misspellings of the pharmacologic inhibitors PD98059 and sp600125 as “PD98509" (including in Fig. 2) and “sp60025” (in the Methods).

Author Response

Dear Reviewer 2

Thank you very much for your reviewing and comments. For your comments and suggestion, please see attached file.

Sincerely,
Prof. Akira Kanda, MD, PhD

Round 2

Reviewer 1 Report

Comments and Suggestions for Authors

The authors  improved the manuscript, but still some text modifications are required before its acceptance, to make the outline of the manuscript understandable for readers that are not familiar with this topic.

Reply 1: In our previous study (Allergol Int. 2020 Apr;69(2):232-238.), we showed that activated tissue eosinophils from nasal polyp of patient with eosinophilic chronic rhinosinusitis (ECRS) accompany with asthma were very highly expressed CD69. Since CD69+ eosinophils can be usually purified no more than 100,000 cells in the one sample, we performed in vitro experiment using mouse with hypereosinophilia, named IL-5 transgenic mouse (Yun Y et al. Allergol Int. 2020 Apr;69(2):232-238. doi: 10.1016/j.alit.2019.11.002). We already showed that almost 15% CD69 expression was observed on the eosinophils from these mice. Of note, previous reports already have shown that level of CD69 is dependent of IL-5 (Immunology. 1990 Oct;71(2):258- 65. and J Leukoc Biol. 2001 Jan;69(1):105-12.). Similarly, we suggested that CD69 expression on eosinophils was dependent on IL-5 concentration (Fig.S3). Then, we used these isolated eosinophils from IL-5Tg mice in our experiment.  

This paragraph should be added in the manuscript for the above described reason, thus enforcing the choice to study all the functional effect only in naïve IL-5Tg mice model.

Probably the authors ignore how to represent statistical differences in the graphs. Please move * on the middle bar in Fig 2E and 2F

Comments on the Quality of English Language

Check typos (see line 93)

Author Response

Dear Reviewer 1

We wish to express our appreciation to the you for insightful comments, which have helped us significantly improve the paper. For reply to second comments, please see attached file.

Yours sincerely,
Akira KANDA

Reviewer 2 Report

Comments and Suggestions for Authors

Revisions to the manuscript have addressed most of this reviewer’s criticisms; however, certain issues remain unaddressed:

1.       The response to Comment 2 does not address the first point: The manuscript claims that Erk1/2 and JNK phosphorylation mediate the effects on IL-10 production and reduced eosinophil viability (e.g., on lines 225-226 of the Discussion). However, data demonstrating that PD98059 does not impact cell death and that sp600125 does not impact IL-10 production are lacking. These data should either be included in the manuscript or this conclusion should be modified to reflect the available data.

2.       As the authors note in their response to Comment 4, PD98059 is an inhibitor of MEK. References to PD98059 as an Erk1/2 inhibitor in the manuscript should be modified accordingly.

3.       Figure 2F notes the concentration of sp600125 as 50 nM, which does not match the figure legend.

Author Response

Dear Reviewer 2

Thank you once again for your valuable comments and suggestions. We are hopeful that our supplementary analyses and revised focus helps to improve your opinion of work.  For reply to your second comments, please see attached file.

Yours sincerely,
Akira KANDA

Round 3

Reviewer 2 Report

Comments and Suggestions for Authors

All reviewer comments have been satisfactorily addressed.